# Estimation of Soluble Solids for Stone Fruit Varieties Based on Near-Infrared Spectra Using Machine Learning Techniques

**DOI:** 10.3390/s22166081

**Published:** 2022-08-14

**Authors:** Pedro Escárate, Gonzalo Farias, Paulina Naranjo, Juan Pablo Zoffoli

**Affiliations:** 1Escuela de Ingeniería Eléctrica, Facultad de Ingeniería, Pontificia Universidad Católica de Valparaíso, Valparaiso 2374631, Chile; 2Departamento de Fruticultura y Enología, Facultad de Agronomía e Ingeniería Forestal, Pontificia Universidad Católica de Chile, Santiago 8331150, Chile

**Keywords:** stone fruits, fruit quality, soluble solids, near infrared spectra, visible spectra, convolutional neural networks, feedforward neural netwoks, classification, absorbance

## Abstract

The quality control for fruit maturity inspection is a key issue in fruit packaging and international trade. The quantification of Soluble Solids (SS) in fruits gives a good approximation of the total sugar concentration at the ripe stage, and on the other hand, SS alone or in combination with acidity is highly related to the acceptability of the fruit by consumers. The non-destructive analysis based on Visible (VIS) and Near-Infrared (NIR) spectroscopy has become a popular technique for the assessment of fruit quality. To improve the accuracy of fruit maturity inspection, VIS–NIR spectra models based on machine learning techniques are proposed for the non-destructive evaluation of soluble solids in considering a range of variations associated with varieties of stones fruit species (peach, nectarine, and plum). In this work, we propose a novel approach based on a Convolutional Neural Network (CNN) for the classification of the fruits into species and then a Feedforward Neural Network (FNN) to extract the information of VIS–NIR spectra to estimate the SS content of the fruit associated to several varieties. A classification accuracy of 98.9% was obtained for the CNN classification model and a correlation coefficient of $R_{c}>0.7109$ for the SS estimation of the FNN models was obtained. The results reported show the potential of this method for a fast and on-line classification of fruits and estimation of SS concentration.

## 1. Introduction

Fruit quality inspection is an important issue in the international trade of fruits and vegetables. Consumer acceptance of fruits [1,2,3,4,5], without the expression of physiological disorder symptoms, is further determined by sugar, acid concentration, and the ratio between both parameters. The quantification of SS in stone fruits gives a good approximation of total sugar concentration at the ripe stage because the sugars comprise the major component of SS and not much storage reserve carbohydrates remain in the fruit. Stone fruits comprise several species with a wide range of varieties expressing different flavors and external and internal quality attributes. Among them, peaches (*Prunus persica* (L.) *Batsch*) and the mutation nectarines and plums (*Prunus salicina* L.) are extensively planted worldwide. Among the quality parameters that determine the eating acceptability of stone fruit, flavor is a complex attribute made up of a mixture of sugars, acids, and volatile components. In peaches, the acceptability of the fruit is strongly related to the concentration of soluble solids and the acidity of the fruit, thus a minimum between 11 and 12% of SS is required [4]. However, there are varieties that can reach SS values above 20% [5]. In plums, a great diversity of varieties have been introduced with different colors and a diversity of flavors and range of soluble solids. Plums with SS higher than 12% show consumer acceptance higher than 75% [1]. Plumcot or pluot are interspecific hybrids, crosses between plum and apricot with notable flavor characteristics [3]. The differences in the values of SS indicate the need to develop SS models for each variety. However, the similarities between the varieties make the visual identification a difficult task; therefore, a method that allows the correct identification of the variety from the VIS–NIR spectra is necessary.

The Quantification of SS using a refractrometer is a well-established procedure to determine the sugar concentration in the fruits [6]. Although the procedure is well diffused in the industry and it has a good relation with the sweetness of the fruit, it cannot always can be performed individually. An average of juice samples, including many fruits, is recorded and used to characterize the population. Therefore a non-invasive and fast technology is required to better understand the individual eating quality of fruits. The non-destructive analysis based on NIR spectroscopy has become a popular technique for the assessment of fruit and vegetables [7,8,9,10] and has been used for estimation of SS concentration in fruits, including apples [6], kiwifruit [11], peaches [12], and nectarines [13] and has been used for tobacco classification [14], fruit classification [15], and mango classification [16]. Moreover, machine learning techniques have been used [16,17,18,19].

In this work, a novel approach based on a CNN and an FNN was developed, to improve stone fruit quality inspection. The CNN was used to classify the fruits into species, and then the FNN was used to estimate the SS concentration among varieties of each species from the VIS–NIR spectra classified. A classification accuracy of 98.9% was obtained for the CNN classification model, and a $R_{c}>0.7109$ was obtained for the SS estimation FNN model.

## 2. Materials and Methods

### 2.1. Samples

A total of 1780 samples of fruits (Table 1) were collected during January and February 2013 from the Central region of Chile (O’Higgins Region). The samples were initially analyzed using an interactance optical setup (Figure 1) to record their absorbance spectra, then a PAL-1, ATAGO Co., Ltd. (Tokio, Japan) equipment was used to process the samples and to measure the soluble solids concentration so that they could be used as reference for the subsequent calibration process.

### 2.2. Hardware

During this work, a spectrometer HR4000TM from Ocean Optics Inc. (Orlando, FL, USA) was used to acquire an individual spectra of the fruit. It incorporates a CCD detector with a linear array of 3648 elements and a diffraction grid (600 lines mm${}^{-1}$ in the 200–1100 nm range), giving a resolution of 0.02–8.4 nm FWHM and a signal-to-noise ratio of 300:1 for wavelengths. The light source is a 150 W EKE Quartz Halogen of 3200 K color temperature.

### 2.3. Optical Design

A diffuse interactance setup was used to reduce the effect of the variability of the sample size. The optical configuration used generates diffuse interactance resulting from a bottom illumination and a 45° lateral reading (Figure 1). A large aperture collimator lens with a field of view of 45° was used to capture high optical power into the spectrometer fiber.

As observed in Figure 1, the light that interacts with the sample is collected and focused by the collimator lens. The emission of the sample is focused into the optical fiber; finally, the spectrometer receives the light and generates the spectrum. The spectrum is transmitted using a USB port into the computer for mathematical processing. A special design for illumination and reading was required. The reason for this design was to avoid reflections and the environmental light being collected by the system.

### 2.4. Acquisition

In the acquisition of the spectra, the absorbance spectra ($A_{\lambda }$) was calculated:(1)$$A_{\lambda }=-log_{10}\frac{S_{\lambda }-D_{\lambda }}{R_{\lambda }-D_{\lambda }},$$
where $S_{\lambda }$ is the emission spectrum, $D_{\lambda }$ is the dark spectrum, and $R_{\lambda }$ is the reference spectrum. The dark spectrum is required to normalize the data, reducing the effect of temperature variations on the CCD. The reference spectrum compensates for illumination instabilities, such as dust, power source variations, and the like. The integration time for the sample spectrum and reference spectrum (150 ms) was defined to ensure that $S_{\lambda }>D_{\lambda }$ and $R_{\lambda }>D_{\lambda }$, for every wavelength so that $A_{\lambda }$ is a real number (Equation (Equation 1)).

Two measures of spectra, one on each side in the equatorial region of the 1780 samples of fruits were performed, giving a total of 3560 spectra. Prior to acquiring the spectra for every fruit, dark and reference spectra were taken. The dark spectrum is obtained in the absence of light for an integration time of 150 ms, and the reference spectrum is obtained positioning an Ocean Optics Inc. WS-1 Diffuse Reflectance Standard in the place of the fruit for an integration time of 150 ms. To guarantee the thermal stabilization of the light source, it was turned on for at least 5 min. to acquire the reference spectra.

The set of samples were randomly separated for calibration (60%), validation (20%), and test (20%). The calibration set was used to train the prediction models; the validation and test set was used to measure the performance of the prediction models to new fruits not included in the calibration set. As mentioned before, the reference measurements for soluble solids concentration were obtained by traditional chemical procedures using a Digital Refractrometer (PAL-1, ATAGO Co., Ltd. (Tokio, Japan)). Summary values are given in Table 2.

### 2.5. Spectral Processing

Due to problems, such as noise in the dark spectra and the noise in the mathematical calculation of the absorbance spectra, it was necessary to apply smoothing techniques to improve the signal-to-noise ratio and improve the prediction performance of the models.

#### 2.5.1. Spectral Correction

To reduce the noise of the absorbance spectra, the Standard Normal Variance [20] and Multiplicative Scattering Correction [21] algorithms were applied to the raw spectra without a positive impact on the results.

#### 2.5.2. Smoothing

To reduce the noise in the absorbance spectra, a first-order Savitsky–Golay filter with a window width between 11–251 was applied to smooth the absorbance spectra. However, the best results were obtained with a window width of 11. Figure 2 shows the spectra of a fruit sample after the Savitsky–Golay filter smoothing process.

### 2.6. Convolutional Neural Network (CNN)

A CNN is a machine learning model widely used as a classification model and recently used in the classification of spectra [8,19]. The main structure of a CNN includes an input layer, a convolution layer, a pooling layer, a fully connected layer, and finally an output layer [22]. In this work, a CNN based on a residual network architecture (ResNet) [23,24,25,26] with a bottleneck block (Figure 3) was used to classify the fruits based on their VIS–NIR absorbance spectra.

#### 2.6.1. Convolutional Layer

The convolutional layer is one of the main blocks of a CNN. In this layer, a dot product operation is performed between the input and a sliding filter(convolutional kernel). Then a bias term is added as is shown in Figure 4.

#### 2.6.2. Batch Normalization Layer

The batch normalization layer is described in [27] and is widely used to accelerate the training process and to make the training process independent of initialization values of the CNN.

#### 2.6.3. ReLU layer

The rectified linear activation function or ReLU is a linear function that will output the input directly if it is positive; otherwise, it will output zero (Equation (Equation 2))
(2)f(x)=max(0,x),

The ReLU layer has become the default activation function used in many types of neural networks [28].

#### 2.6.4. Pooling Layer

The pooling process is very similar to the convolution operation. A filter of length N is slid over the input to calculate an output. The pooling layer is usually used to down-sample the input and to reduce the quantity of parameters to estimate in the CNN [29]. There are several functions used as pooling layers. The most commonly used functions are the max pooling (Equation (Equation 3)) and the average pooling (Equation (Equation 4)).
(3)$$c_{j}=max\left(x_{j}^{\prime },x_{j+1}^{\prime },\ldots ,x_{j+N-1}^{\prime }\right),$$
(4)$$c_{j}=\left(x_{j}^{\prime }+x_{j+1}^{\prime }+\ldots +x_{j+N-1}^{\prime }\right)/N,$$

#### 2.6.5. Fully Connected Layer

The fully connected layer corresponds to a FNN. In this work, a fully connected layer with 6 outputs was used.

#### 2.6.6. Softmax Layer

The softmax function is commonly used in a classification process [30]. The softmax value of an array X is calculated by:(5)$$S_{i}=\frac{e^{x_{i}}}{\sum_{j=1}^{N}e^{x_{j}}}$$

In a CNN, the softmax function is used as a measure of the probability that the sample $x_{i}$ belongs to a specific category.

### 2.7. Feedforward Neural Netwok (FNN)

An FNN is a kind of neural network which consists of an input layer, a hidden layer, and an output layer. The standard structure of an FNN includes an input layer, a hidden layer, and an output layer of 1 neuron (Figure 5).

Finally, an FNN using the architecture of Figure 5 was used to estimate the soluble content of the fruits based on their VIS–NIR spectra.

### 2.8. Model Training Parameters

Each layer of the CNN and FNN have parameters that affect the training process. The detail of the parameters and the values for the training process are described in Table 3.

### 2.9. Model Performance Evaluation

To evaluate the performance of the trained CNN model, the accuracy index that indicates the rate of correctly classified samples, was calculated by:(6)$$Accuracy=\frac{TP+TN}{N_{T}}*100\%,$$
where a true positive (TP) and a true negative (TN) indicate an accurate identification, and $N_{T}$ represents the total number of samples.

The performance of the SS FNN models was assessed in terms of the Root Mean Square Error of the calibration set (RMSEC), the Root Mean Square Error of the validation set (RMSEV), the Root Mean Square Error of the test set (RMSET) and the correlation coefficient (*R*). These values are given by:(7)$$RMSE\left(C/V/T\right)=\sqrt{\frac{\sum_{i=1}^{N}\left({\hat{y}}_{i}-y_{i}\right)}{N-1}},$$
(8)$$R_{C/V/T}=\frac{Cov({\hat{y}}_{i},y_{i})}{\sqrt{\sigma_{y}\sigma_{\hat{y}}}},$$
where $y_{i}$ and ${\hat{y}}_{i}$ are the reference and predicted values of the sample, respectively.

## 3. Results and Discussion

Eight neural networks models were trained, one CNN for classification of the fruit by species (Plumcot (PL), Peaches (PE), Black Plums (BP), Red Plums (RP), White pulp nectarines (WPN), and Yellow pulp nectarines (YPN)), one FNN soluble solids model for each fruit species defined in Table 1, and one FNN soluble solids including all samples. The aim was to find the best soluble solid FNN model compared with the reference data obtained from the Digital Refractometer. For the training of the classification CNN model, the spectra were separated in two groups: calibration (70%, 2492 samples) and validation (30%, 1068 samples) The group of spectra (Figure 2) for each FNN SS model was randomly separated in groups: calibration (60%, 1536 samples), validation (20%, 712 samples), and test (20%, 712 samples). The calibration set was used to train the models, and the validation and test set was used to measure the performance of the models to classify or estimate the SS values of new fruits not included in the calibration set. To guarantee thermal stability for the different setups, the excitation sources were put to work at least 5 min prior to any measurement. The results obtained for the classification CNN model are summarized in Figure 6.

The results show that the CNN classification model has an accuracy greater than 97.1%. Only 12 samples were incorrectly classified in the validation set: 2 BP classified as PL, 1 PE classified as WPN, 1 PL classified as BP, 4 WPN classified as YPN, and 2 YPN classified as WPN. The results show that spectra contain useful information about the classification of fruit species, and it is possible to correctly classify the fruit 98.9% of the time. Due to this performance of the classification model, a specific SS FNN model for each fruit was trained. The calibration, validation, and test results obtained for the SS FNN models are summarized in Table 4.

The results from Table 4 show that the SS FNN RP model has the best performance in terms of the RMSE and *R*. On the other hand, the YPN has the worst performance in terms of the RMSE and *R*. In addition, the results show that a model including all samples has a lower performance in terms of RMSE. This lower performance demonstrates the need to have a model that allows the classification of fruits according to their spectrum to apply the correct SS FNN model to each species of fruit. Even though the spectra (Figure 2) show two noisy regions, it was possible to extract useful information from the VIS–NIR spectra to train the classification and SS models. Nevertheless, in a future work an improvement in the acquisition of the absorbance spectra is necessary to reduce the sources of instability.

## 4. Conclusions

In this work, we addressed the estimation of soluble solids in stone fruits using VIS–NIR spectra. We proposed the use of a classification model before the soluble solid estimation model. The results obtained demonstrate that it is possible to use machine learning techniques to develop a classification model that identifies the correct SS model to use for on-line determination of fruit maturity and quality control. This is an important result for the quality control issue because it allows reducing the error of choosing an incorrect model or using a general model that has a lower performance than a specific model.

## Figures and Tables

**Figure 1 sensors-22-06081-f001:**
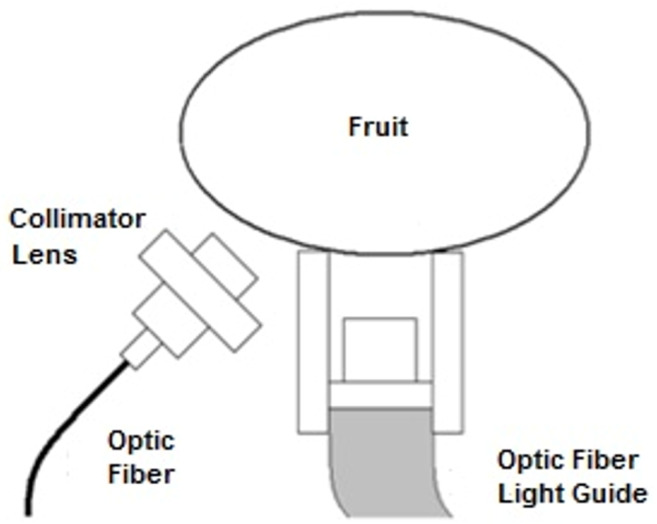
Optics setup.

**Figure 2 sensors-22-06081-f002:**
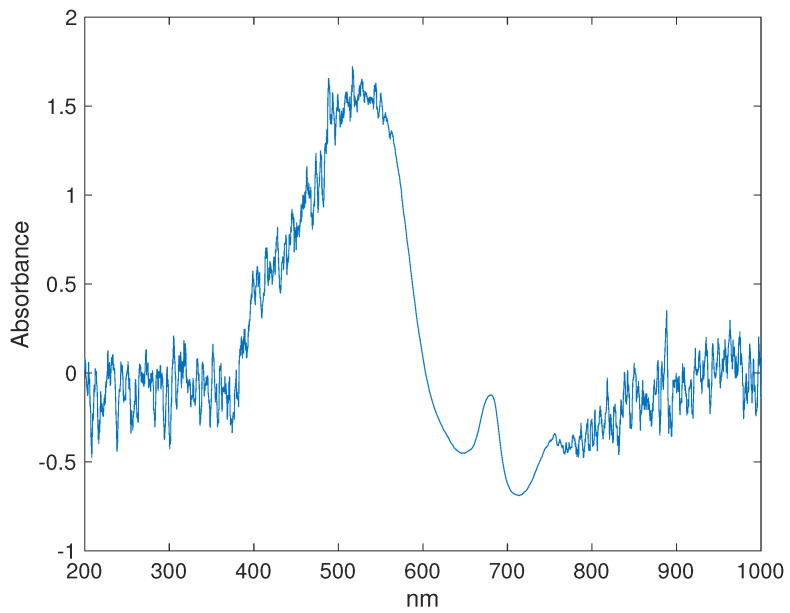
Absorbance Spectra of Beauty Sweet peach.

**Figure 3 sensors-22-06081-f003:**
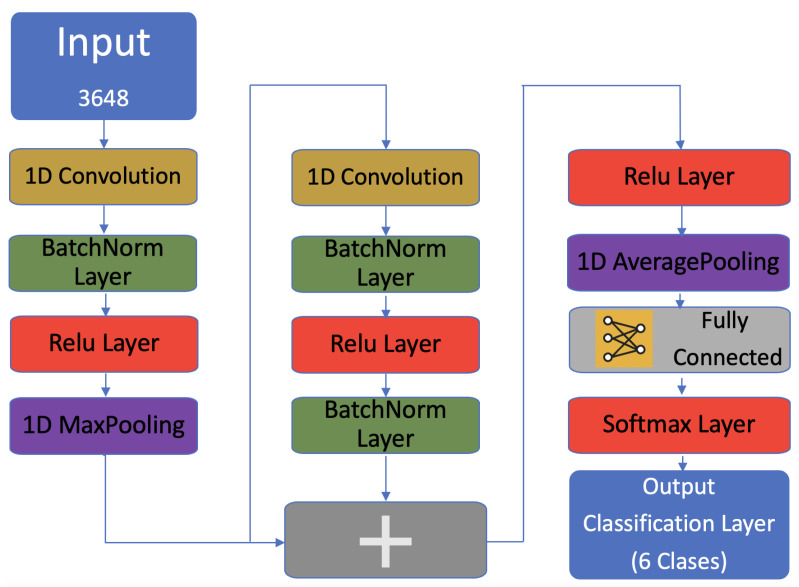
ResNet architecture.

**Figure 4 sensors-22-06081-f004:**
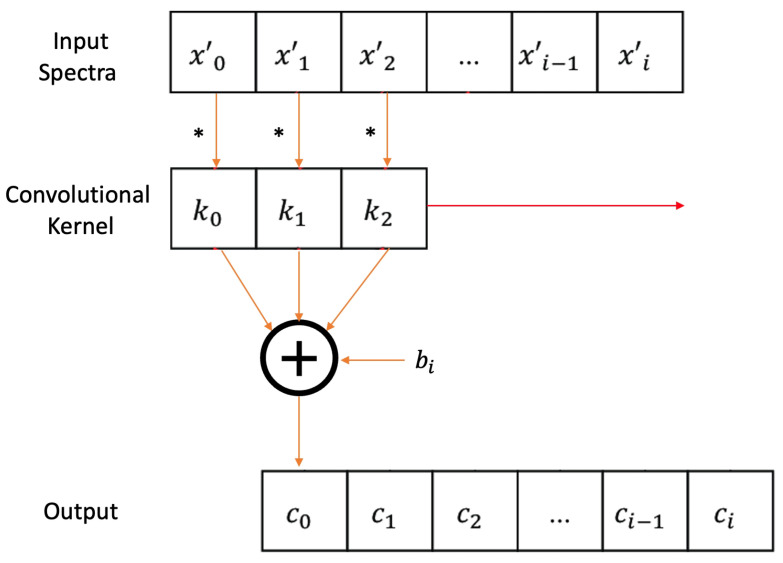
Schematic diagram of 1D convolution operation.

**Figure 5 sensors-22-06081-f005:**
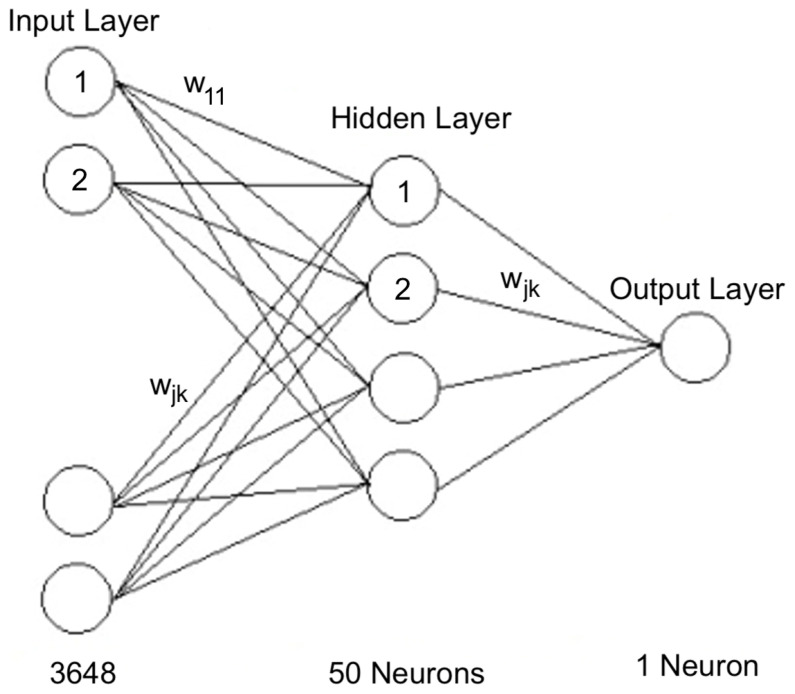
FNN architecture. $w_{jk}$ represent the weight of the connection.

**Figure 6 sensors-22-06081-f006:**
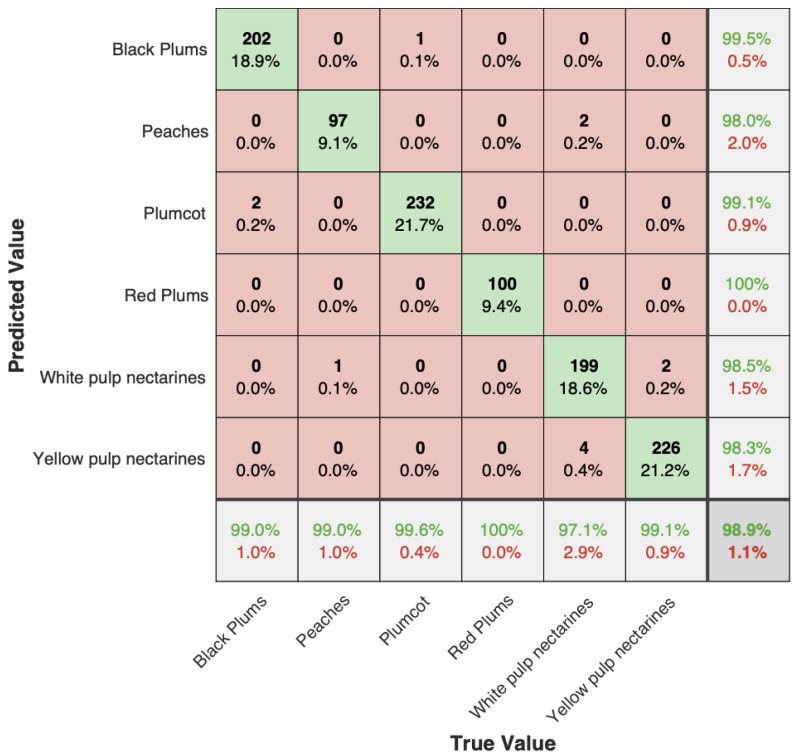
Confusion matrix of the validation group. The green squares represent the number of TP samples, the light pink squares represent the number of samples that were incorrectly classified, the green values represent the percentage of TP samples and the red values represent the percentage of samples that were incorrectly classified.

**Table 1 sensors-22-06081-t001:** Number of samples by species of stone fruits.

Fruit Species	Variety	Number of Samples
Peaches	Beauty Sweet	40
	Elegant Lady	60
	September Sun	60
	Zee Lady	20
Yellow Pulp Nectarines	Ruby Diamond	20
	Summer Diamond	40
	Red Jim	100
	Zee Glo	60
	Venus	60
	August Red	80
White Pulp Nectarines	Arctic Snow	60
	August Pearl	140
	Giant Pearl	140
Red Plums	Fortune	80
	Red Heart	100
Black Plums	Angeleno	80
	Autumn Pride	120
	Black Kat	120
Plumcots	Blue Gusto	120
	Dapple Dandy	80
	Flavor Granade	120
	Flavor Rich	80

**Table 2 sensors-22-06081-t002:** Soluble solids reference values statistics.

	Min.	Max	Mean	Std. Dev.
Soluble Solids Concentration (%)	6.3	20.9	12.29	2.57

**Table 3 sensors-22-06081-t003:** Model training parameters.

Model	Layer	Parameters
CNN	Convolution 1	Filter Size: 7 Number Filters: 64
	BatchNorm 1	Mean Decay: 0.1 Variance Decay: 0.1 Epsilon: 0.00001
	Max Pooling	Pool Size: 5 Stride: 1
	Convolution 2	Filter Size: 3 Number Filters: 64
	BatchNorm 2	Mean Decay: 0.1 Variance Decay: 0.1 Epsilon: 0.00001
	Convolution 3	Filter Size: 3 Number Filters: 64
	BatchNorm 3	Mean Decay: 0.1 Variance Decay: 0.1; Epsilon: 0.00001
	Average Pooling	Pool Size: 5 Stride: 1
	Fully Connected	Output Size: 6
Training Algorithm: Stochastic gradient descent with momentum (SGDM)		
Learning Rate: 0.0001; Epochs: 200		
FNN	Hidden	Neurons: 50
Training Algorithm: Scaled Conjugate Gradient		
Epochs: 200		

**Table 4 sensors-22-06081-t004:** Soluble solids models performance.

Model	RMSEC	RMSEV	RMSET	$R_{c}$	$R_{v}$	$R_{T}$
RP	0.6303	0.7054	0.5823	0.9206	0.8693	0.9548
BP	0.7345	0.6707	0.6444	0.9106	0.9171	0.9363
PE	1.0935	0.9225	1.1739	0.8408	0.9027	0.8033
YPN	1.8634	1.5861	1.9085	0.7109	0.7448	0.6681
WPN	1.3519	0.9225	1.1625	0.8981	0.9533	0.9306
PL	1.2688	0.8454	1.0859	0.8693	0.9407	0.9123
All	1.4108	1.4384	1.4650	0.8340	0.8258	0.8237

## Data Availability

Not applicable. No new data were created or analysed in this study. Data sharing is not applicable to this article.

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
