# Peer review of "Estimation of Soluble Solids for Stone Fruit Varieties Based on Near-Infrared Spectra Using Machine Learning Techniques"

_sensors, 2022, doi:10.3390/s22166081_

Round 1

Reviewer 1 Report

1.      Please enrich the information in the introduction section, it is too short and cannot give enough background information.

2.      Figure 2, it is noisy even after SG preprocessing, more preprocessing techniques should be used to reduce the noises. It is quite important.

3.      There is no need to introduce CNN in detail

4.      Please try to combine all types of fruits for regression

5.      In depth discussion should be conducted for this study and related studies

Author Response

We thank to Reviewer for all important comments regarding our paper. We provide a detailed re- sponse to the comments in order to improve the current presentation of the manuscript:

1) Please enrich the information in the introduction section, it is too short and cannot give enough background information.

We thank the reviewer for this comment. We include a paragraph with background information in the introduction.

2) Figure 2, it is noisy even after SG preprocessing, more preprocessing techniques should be used to reduce the noises. It is quite important.

We thank the reviewer for this comment. Perhaps we were not clear enough on that point. We tried several preprocessing techniques without a positive impact on the results. For clarity of the presentation, we include information about the preprocessing techniques in the section 2.5. ”Spectral Processing”. Also we add a paragraph in lines 204-208.

3) There is no need to introduce CNN in detail.

We thank the reviewer for this comment. For clarity of the presentation, due to it is possible to find in the literature CNNs with different architectures, we decided to include a detailed description of the CNN layers and their parameters in this work.

4) Please try to combine all types of fruits for regression

We thank the reviewer for this comment. The table 4 shows the performance of a model including all samples. For clarity of the presentation we change the paragraph in the page 10 (lines 165-167) ”The results from the Table 4 show .... coefficient R.”

5) In depth discussion should be conducted for this study and related studies

We thank the reviewer for this comment. For clarity of the presentation of the state of the art in machine learning applied to NIR data, we made some changes in the introduction. In the page 2, lines 47-50, there are references of SS estimation in fruits and vegetables [1,2,5,6], references of classification of fruits and vegetables using NIR spectra [3,4,7] and finally references with the use of machine learning [7–10]. We estimate that the references in the introduction provide a theoretical background and a detailed description of related studies.

References

  1. [1]  G. Carlomagno, L. Capozzo, G. Attolico, and A. Distante. Non-destructive grading of peaches by near-infrared spectrometry. Infrared Physics Technology, 46(1):23–29, 2004. Workshop on Advanced Infrared Technology and Application.

  2. [2]  Carlos H. Crisosto. Stone fruit maturity indices: a descriptive review. 1994.

  3. [3]  Marcelo C. A. Marcelo, Frederico L. F. Soares, Jorge A. Ardila, Jailson C. Dias, Ricardo Ped ́o, Samuel Kaiser, Oscar F. S. Pontes, Carlos E. Pulcinelli, and Guilherme P. Sabin. Fast inline to- bacco classification by near-infrared hyperspectral imaging and support vector machine-discriminant analysis. Anal. Methods, 11:1966–1975, 2019.

  4. [4]  Nishtha Parashar, Aman Mishra, and Yatin Mishra. Fruits classification and grading using vgg-16 approach. In Vishal Goyal, Manish Gupta, Seyedali Mirjalili, and Aditya Trivedi, editors, Proceedings of International Conference on Communication and Artificial Intelligence, pages 379–387, Singapore, 2022. Springer Nature Singapore.

  5. [5]  David C. Slaughter and Carlos H. Crisosto. Nondestructive internal quality assessment of kiwifruit using near-infrared spectroscopy. In Seminars in food analysis, volume 3, pages 131–140. CHAPMAN & HALL, 1998.

  6. [6]  DC Slaughter. Nondestructive determination of internal quality in peaches and nectarines. Trans- actions of the ASAE, 38(2):617–623, 1995.

  7. [7]  Susanto, Suroso, I. Wayan Budiastra, and Hadi K. Punvadaria. Classification of mango by artificial neural network based on near infrared diffuse reflectance. IFAC Proceedings Volumes, 33(29):157– 161, 2000. 2nd IFAC/CIGR International Workshop on Bio-Robotics, Information Technology and Intelligent Control for Bioproduction Systems (BIO-ROBOTICS II), Sakai, Osaka, Japan, 25-26 November 2000.

  8. [8]  Di Wang, Fengchun Tian, Simon X. Yang, Zhiqin Zhu, Daiyu Jiang, and Bin Cai. Improved deep cnn with parameter initialization for data analysis of near-infrared spectroscopy sensors. Sensors, 20(3), 2020.

  9. [9]  Rongda Xu, Lei Zhang, Xiangqian Ding, and Ruichun Hou. Classification modeling method for near-infrared spectroscopy of tobacco based on multimodal convolution neural networks. Journal of Analytical Methods in Chemistry, 2020:9652470, 2020.

Corresponding author: P. Esc ́arate [email protected] 2

Paper review: Sensors-1786617

[10] Xiaolei Zhang, Jinfan Xu, Jie Yang, Li Chen, Haibo Zhou, Xiangjiang Liu, Haifeng Li, Tao Lin, and Yibin Ying. Understanding the learning mechanism of convolutional neural networks in spectral analysis. Analytica Chimica Acta, 1119:41–51, 2020.

Reviewer 2 Report

The manuscript entitled ‘Estimation of soluble solids based on near-infrared spectra using machine learning techniques’ by Escarate et al. is interesting and a better contribution to the quality control of fruit packing. The manuscript needs some revisions before it can be recommended for publication.

1. All authors’ e-mail should be provided.

2. Page 1, line 2: Does soluble solid have to be capitalized?

3. Page 1, line 5: Near-infrared, N or n? The author should define the abbreviation, NIR, here. By the way, remove the NIR next line.

4. Page 1, line 9: Feed Forward, Feedforward or feedforward?

5. Page 1, line 14: The requirement for keywords is 3-10. Please include more keywords.

6. A mild suggestion: The authors have many abbreviations in the manuscript. It is better to have a ‘Abbreviation section’ for readers.

7. Page 1, lines 16-17: The first sentence of Introduction is exactly the same as the first sentence of Abstract. Please rewrite.

8. Page 1, Figure 28: Does ‘Near-infrared’ have to be capitalized?

9. Page 1, Introduction section: There is only one paragraph. A mild suggestion: the authors can separate it into two paragraphs from line 31, in this work…..

10. Page 2, lines 39-40: the authors mentioned 400 samples were collected in 2013. However, in table 1, there are 1680 samples in total, why? Please describe when the experiment performed?

11. Page 2, line 42: Atago, Co (City name, Country name). Please provide the info.

12. The table and figure legends are too simple, please add more details.

13. Page 4, Figure 2: Absorbance spectra of what fruit sample (line 89)?

14. Page 4, Table 2: Std. Desv. is Spanish not English.

15.Page 5, line 97: Arquitecture is a typo of architecture. Please also correct the same typo in Figure 5.

16. Page 7, line 128: w of where should be capitalized.

17. Page 7, line 131-132: in Table . 3 is incorrect. Please give more details about the parameters.

18. Page 9, lines 155-156: Figure 6), ) is unnecessary.

19. Page 10, Conclusions section is alike Result and discussion. Please consider rewrite.

20. Please provide the following info: Acknowledgement, Funding, Data Availability Statement, Conflicts of Interest…etc.

Author Response

We thank to Reviewer for all important comments regarding our paper. We provide a detailed re- sponse to the comments in order to improve the current presentation of the manuscript:

1) The manuscript entitled ‘Estimation of soluble solids based on near-infrared spectra using machine learning techniques’ by Escarate et al. is interesting and a better contribu- tion to the quality control of fruit packing. The manuscript needs some revisions before it can be recommended for publication.

We thanks the reviewer for this comment. For clarity of the presentation, we made some changes in the manuscrpt.

2) All authors’ e-mail should be provided.

We include the e-mail in the affiliation.

3) Page 1, line 2: Does soluble solid have to be capitalized?

Done.

4) Page 1, line 5: Near-infrared, N or n? The author should define the abbreviation, NIR, here. By the way, remove the NIR next line..

Done.

5) Page 1, line 9: Feed Forward, Feedforward or feedforward?

We thanks the reviewer for this comment. We changed Feed Forward by Feedforward.

6) Page 1, line 14: The requirement for keywords is 3-10. Please include more keywords.

Done.

7) A mild suggestion: The authors have many abbreviations in the manuscript. It is better to have a ‘Abbreviation section’ for readers..

We thanks the reviewer for this comment. We include an Abbreviations section.

8) Page 1, lines 16-17: The first sentence of Introduction is exactly the same as the first sentence of Abstract. Please rewrite.

We thanks the reviewer for this comment. We rewrite the introduction.

9) Page 1, Figure 28: Does ‘Near-infrared’ have to be capitalized?

Done.

10) Page 1, Introduction section: There is only one paragraph. A mild suggestion: the authors can separate it into two paragraphs from line 31, in this work......

Done.

11) Page 2, lines 39-40: the authors mentioned 400 samples were collected in 2013. How- ever, in table 1, there are 1680 samples in total, why? Please describe when the experiment performed?.

We thanks the reviewer for this comment. There are 1780 samples of fruits (we change the number in the manuscript), also we change the paragraph ”Two measures ... spectra” in section Acquisition for ”Two measures of spectra, one on each side in the equatorial region of the 1780 samples of fruits were performed giving a total of 3560 spectra”.

12) Page 2, line 42: Atago, Co (City name, Country name). Please provide the info..

We add the data of the Headquaters of ATAGO CO, LTD.

13) The table and figure legends are too simple, please add more details..

Done.

14) Page 4, Figure 2: Absorbance spectra of what fruit sample (line 89)?.

We add the description in the Figure 2.

15) Page 4, Table 2: Std. Desv. is Spanish not English.

Done.

16) Page 5, line 97: Arquitecture is a typo of architecture. Please also correct the same typo in Figure 5..

Done.

17) Page 7, line 128: w of where should be capitalized.

Done.

18) Page 7, line 131-132: in Table . 3 is incorrect. Please give more details about the parameters.

We thanks the reviewer for this comment. We change the paragraph.

19) Page 9, lines 155-156: Figure 6), ) is unnecessary.

Done.

20) Page 10, Conclusions section is alike Result and discussion. Please consider rewrite..

We thanks the reviewer for this comment. We made changes in the Results and discussion section and Conclusions

21) Please provide the following info: Acknowledgement, Funding, Data Availability Statement, Conflicts of Interest... etc..

Done.

Round 2

Reviewer 1 Report

The authors have addressed my comments. 

Reviewer 2 Report

Some typos can be corrected by scientific editor. The authors responsed all my comments. To me, it is acceptable!